# Very Low Alcohol Consumption Is Associated with Lower Prevalence of Cirrhosis and Hepatocellular Carcinoma in Patients with Non-Alcoholic Fatty Liver Disease

**DOI:** 10.3390/nu14122493

**Published:** 2022-06-16

**Authors:** Silvia Ferri, Bernardo Stefanini, Lorenzo Mulazzani, Margherita Alvisi, Francesco Tovoli, Simona Leoni, Luca Muratori, Tommaso Lotti, Alessandro Granito, Luigi Bolondi, Fabio Piscaglia

**Affiliations:** 1Division of Internal Medicine, Hepatobiliary and Immunoallergic Diseases, IRCCS Azienda Ospedaliero-Universitaria di Bologna, 40138 Bologna, Italy; bernardo.stefanini@studio.unibo.it (B.S.); lorenzo.mulazzani2@studio.unibo.it (L.M.); margherita.alvisi@studio.unibo.it (M.A.); francesco.tovoli2@unibo.it (F.T.); simona.leoni@aosp.bo.it (S.L.); luca.muratori@studio.unibo.it (L.M.); alessandro.granito@unibo.it (A.G.); fabio.piscaglia@unibo.it (F.P.); 2Department of Medical and Surgical Sciences, University of Bologna, 40126 Bologna, Italy; lottitommaso.med@gmail.com (T.L.); luigi.bolondi@unibo.it (L.B.)

**Keywords:** NAFLD, alcohol, wine, cirrhosis, hepatocellular carcinoma

## Abstract

The role of moderate alcohol consumption in the evolution of NAFLD is still debated. The aim of this study is to evaluate the impact of current and lifelong alcohol consumption in patients with NAFLD. From 2015 to 2020, we enrolled 276 consecutive patients fulfilling criteria of NAFLD (alcohol consumption up to 140 g/week for women and 210 g/week for men). According to their current alcohol intake per week, patients were divided in: abstainers, very low consumers (C1: <70 g/week) and moderate consumers (C2). We created a new tool, called LACU (Lifetime Alcohol Consuming Unit) to estimate the alcohol exposure across lifetime: 1 LACU was defined as 7 alcohol units per week for 1 drinking year. Patients were divided into lifelong abstainers and consumers and the latter furtherly divided into quartiles: Q1-Q4. Stratification according to alcohol intake, both current and cumulative as estimated by LACU, showed that very low consumers (C1 and Q1-Q3) displayed lower frequency of cirrhosis and hepatocellular carcinoma compared to abstainers and moderate consumers (C2 and Q4). We can speculate that up to one glass of wine daily in the context of a Mediterranean diet may be a long-term useful approach in selected NAFLD patients.

## 1. Introduction

Non-alcoholic fatty liver disease (NAFLD) is characterised by excessive hepatic fat accumulation, defined as the presence of steatosis in >5% of hepatocytes, detectable by imaging techniques or histology. NAFLD includes a spectrum of disorders ranging from simple fatty liver (NAFL) to non-alcoholic steatohepatitis (NASH) including fibrosis, cirrhosis and hepatocellular carcinoma (HCC). Obesity, insulin-resistance or type 2 diabetes mellitus, arterial hypertension and dyslipidemia, key elements of the so-called metabolic syndrome (MS), are the most relevant conditions related to NAFLD [1]. The diagnosis of NAFLD requires the exclusion of secondary causes of hepatic fat accumulation, in particular the exposition to known steatogenic drugs and an excessive alcohol consumption. There is no generalised agreement in the definition of the dose of alcohol to be considered “harmless” for the liver, especially in presence of cofactors such as NAFLD. The relationship between alcohol and liver injury depends on several cofactors (amount of alcohol consumption, type of alcoholic beverage, drinking patterns, duration of exposure, individual/genetic susceptibility) and patients consuming moderate amounts of alcohol may also be predisposed to NAFLD if they have metabolic risk factors. Several guidelines of national and international Societies for the study of the liver worldwide give different safety thresholds for alcohol consumption [2]: for example EASL (European Association for the Study of the Liver), NICE (National Institute for Health and Care Excellence) and AISF (Italian Association for the Study of the Liver) consider <30 g per day in men and <20 g/day in women [1,3,4]; AASLD (American Association for the Study of Liver Disease) similarly indicates <21 standard drinks per week in men and <14 in women [5]; Asia-Pacific working party on NAFLD considers safe only a lower amount of alcohol, namely <14 standard drinks per week for men and <7 drinks per week in women [6]. Several studies on the role of alcohol consumption in patients with NAFLD have been carried out. While the reports generally agree on the presence of a specific additional risk for HCC provided by the intake of moderate amount of alcohol in NAFLD patients with advanced liver disease [7,8,9], the effect of subthreshold mild degrees of alcohol use in uncomplicated NAFLD is still debated. The reasons leading to inconsistent results include cross-sectional design of most of the studies, occurrence of few liver-related events in longitudinal observations, different instruments for NALFD diagnosis, different definitions of “alcoholic unit” (ranging from 8 to 12 g each), diverse selection criteria due to the guidelines applied, incomplete adjustment for confounders as lifestyle factors and principal focus on current alcohol consumption [10,11]. Actually, most studies on moderate alcohol use in NAFLD evaluate ongoing drinking habits but do not include lifetime drinking histories or patterns which might be more relevant than current alcohol intake. The few studies analysing the role of lifetime alcohol consumption in NAFLD patients used different questionnaire tools to optimize the difficult recall of drinking habits, thus are not fully comparable [12,13,14]. Validated questionnaires for alcohol consumption (as AUDIT or CAGE) have been designed to identify patients at high risk for alcohol use disorders, and are not suitable to differentiate subgroups of patients with NAFLD that are all included in the low risk/risky population [15]. The aim of our study is to evaluate the clinical hepatic impact of current and lifelong moderate alcohol consumption in a cohort of outpatients with NAFLD.

## 2. Materials and Methods

Patients: We enrolled 300 consecutive patients, from 1 March 2015 to 1 February 2020, at the time of their first visit in our tertiary outpatient clinic for liver diseases and liver tumors, fulfilling criteria of NAFLD: fatty liver at ultrasound, HSI (hepatic steatosis index) > 30 and exclusion of other well-known causes of fatty liver (significative alcohol consumption according to EASL guidelines [1], steatogenic drugs and liver storage diseases). Patients with current or previous history of HBV or HCV infections were also excluded. A structured interview about alcohol consumption habits was performed at enrolment by the same physician for all patients. The interview was repeated at a time distance of a median of two years by a trained Medicine Student by telephone call. The interview included questions on current and lifetime alcohol consumption besides pattern and quality of alcoholic beverages to verify the reproducibility and reliability of the reported information about life-long alcohol intake. Of 300 patients, 11 were excluded due to the impossibility to perform the second interview and 13 were excluded due to a poor reliability on lifetime alcohol use, as the answers given in the two abovementioned interviews were greatly different. We thus present data on 276 patients. Patients underwent an accurate physical examination with recording of height, weight and waist measurements without shoes and with light clothing. Arterial blood pressure was recorded after a five-minute resting period. A full physiological and pathological history was recorded. A diagnosis of diabetes mellitus type 2 was assumed if the disease had been previously diagnosed, or in case of history of ongoing or previous pharmacological treatment for diabetes or fasting glucose > 116 mg/dl in at least two consecutive determinations. Insuline-resistance was defined as HOMA (homeostasis model assessment) index > 2.5 in at least two determinations. Arterial hypertension was defined as previously diagnosed, ongoing or previous pharmacological treatment for hypertension or finding of blood pressure > 135/90 mmHg on two separate occasions. Coffee and soft drink servings per week were estimated using a quantitative food frequency questionnaire: one soft drink serving was defined as 200 mL of beverage and one coffee serving was defined as one “espresso” (small coffee cup). Weekly physical activity during the year before the first visit was evaluated using a quantitative questionnaire: up to 30 min of leisure-time physical activity were classified as 1 (scarce), 30–90 min were classified as 2 (moderate) and more than 90 min were classified as 3 (good). Patients were also divided into non-smokers and smokers according to whether had never or ever smoked. For the latter group, exposition to cigarette smoke was quantified in pack-years, defined as twenty cigarettes smoked every day for one year.

Alcohol consumption evaluation: paralleling smoke exposition, we created a new tool, called LACU (Lifetime Alcohol Consuming Unit) to estimate the cumulative amount of alcohol consumed in lifetime by our patients.

We chose to consider weekly instead of daily alcohol use to correctly evaluate social drinkers who usually drink more during the weekends. One alcohol unit was measured as 10 g of pure alcohol, which corresponds to a small glass (125 mL) of lower strength wine (10%), half a pint (280 mL) of lower strength (4%) beer or cider or a single measure (25 mL) of spirit (40%).

Quantitative evaluation:Current use:

Our study included patients who reported a weekly alcohol consumption < 210 g for men and <140 g for women. Patients were divided into 3 groups according to ongoing alcohol use: A (abstainers, < 1 g per week), C1 (low consumers, 1–70 g per week) and C2 (moderate consumers, 71–210 g per week for men and 71–140 g per week for women).

2.Lifetime alcohol consumption units (LACU):

LACU definition: 1 LACU was considered as 7 alcohol units per week for 1 drinking year from teenage onwards. Patients were divided into 2 groups according to lifelong alcohol use: A (abstainers, < 0.1 LACU) and C (consumers). C were further divided into 4 groups using quartile range as Q1 (<4.29 LACU), Q2 (4.30–12.85 LACU), Q3 (12.86–40.00 LACU), Q4 (>40.01 LACU).

Qualitative evaluation:Drinking patterns (binge defined by 5 or more units at one sitting for men and 4 for women, with an overall consumption included into the parameters to define NAFLD as previously reported)Kind of alcoholic beverages consumed (exclusively or preferably wine, beer or spirits)

Fibrosis evaluation: Fibrosis was estimated in all patients using validated non-invasive markers: NAFLD fibrosis score (NFS) [16], FIB-4 [17] and two-dimensional transient elastography (2D-SWE, Hologic Aixplorer SuperSonic Imagine, Aix-en-Provence, France) [18]. Cirrhosis was diagnosed on standard clinical, laboratory, ultrasound, elastosonographic and/or histology examinations. In particular, patients were classified as cirrhotic if they had imaging (ultrasound, computed tomography or magnetic resonance) features of cirrhosis and/or histology or elastosonographic evidence of F4 fibrosis (corresponding to a liver stiffness > 13.0 kPa).

Statistical analysis: Statistical analysis was performed with SPSS version 23.0 (SPSS Inc., Chicago, Il, USA). Categorical data are presented as number (percentage) and were compared using the Chi-square test. Continuous data are presented as median (range) and were compared with Mann-Whitney test. Univariate and multivariate binary logistic regression was carried out to assess odds ratios of variable significantly associated with the risk of cirrhosis and HCC. *p*-values < 0.05 were considered to be statistically significant. In case of multiple pairwise comparisons between groups, multiplicity was addressed adjusting the *p*-value threshold of significance according to the Bonferroni correction for continuous variables and to the Dunn-Bonferroni correction for categorical variables.

## 3. Results

The characteristics of the cohort of 276 patients are reported in Table 1 and are in line with the usual population for Italian NAFLD patients. Patients were overweight (median BMI 29), middle aged (median 59 years), often hypertensive (57.2%) and diabetic (26.8%) or with insulin-resistance (median HOMA 2.3). Prevalence of cirrhosis (20.3%) and hepatocellular carcinoma (11.6%) were higher than in the usual Italian NAFLD population as our unit is a hepatology tertiary referral centre.

### 3.1. Comparison of Patients According to Current Alcohol Intake

Active alcohol consumers were more frequently males (69.0% vs. 37.6%, *p* < 0.001) and a slightly higher proportion of them had been cigarette-smokers (50.5% vs. 38.0%, *p* = 0.06), with a comparable overall toxic exposure between the smokers of the two groups as attested by pack-years calculation (Table 2).

No differences were detected in BMI or frequency of arterial hypertension, dyslipidemia or diabetes though abstainers showed higher levels of glycosylated haemoglobin (48 mmol/mol vs. 41 mmol/mol, *p* = 0.05). No significant differences emerged between the two groups regarding alimentary habits or leisure physical activity apart from a slightly higher coffee consumption in the drinkers group (*p* = 0.04). Regarding laboratory parameters, the only differences were a lower ferritin and a higher HDL cholesterol serum level in abstainers compared to active drinkers (*p* = 0.04 and *p* = 0.03 respectively). No differences were detected in ultrasound parameters or HIS between current abstainers and drinkers. All non-invasive markers of fibrosis (NFS, FIB4, two-dimensional transient elastography) were comparable between abstainers and current drinkers, whereas clinical detection of cirrhosis was more frequent in current abstainers (28.3% vs. 16.3%, *p* = 0.02). HCC was equally represented in the two groups.

We then analysed different current drinkers subgroups: C1 were younger than abstainers (*p* < 0.01) and C2 (*p* < 0.001) while male sex was more represented in C2 (*p* < 0.001 vs. abstainers and C1). No differences were detected in BMI, frequency of arterial hypertension, dyslipidemia or diabetes, although HOMA index in non-diabetic patients was slightly higher in C2 compared to C1 (*p* = 0.02). Smoking exposure, leisure physical activity and alimentary habits were also similar. Exclusive wine and spirits consumption was equally represented in the two drinkers groups whereas beer (both exclusive and overall) was consumed more frequently in C1 and overall wine and spirits were consumed more frequently in C2. A binge-drinking pattern was reported more frequently in C2 (*p* = 0.01). As for laboratory parameters, ferritin was slightly higher in C2 than in C1 and abstainers (*p* = 0.03 for both) and in C1 compared to abstainers (*p* = 0.03). HSI was comparable among groups. Despite similar FIB4 values, NFS was lower in C1 compared to C2 (*p* = 0.01). 2D-SWE was lower in C1 compared to abstainers (*p* = 0.002) and C2 (*p* < 0.001). Accordingly, C1 showed the lowest frequency of cirrhosis (*p* = 0.001 vs. abstainers and *p* = 0.003 vs. C2) and HCC (*p* = 0.009 vs. abstainers and *p* < 0.001 vs. C2). No difference in the prevalence of cirrhosis emerged between abstainers and C2. Despite a lower prevalence of HCC in abstainers compared to C2, the difference did not reach a statistical significance (*p* = 0.08).

Figure 1 shows the prevalence of advanced liver disease in our cohort of patients according to current alcohol consumption.

### 3.2. Comparison of Patients According to LACU Status

Table 3 shows the characteristics of the whole cohort of 276 patients divided between lifetime abstainers and alcohol consumers (including current and past consumers). Alcohol consumers were more frequently males and a higher proportion of them has been a cigarette-smoker, with a comparable overall toxic exposure between the smokers of the two groups according to pack-years estimation. Alcohol abstainers were more often affected by diabetes mellitus (*p* = 0.03) with higher levels of glycosylated haemoglobin (*p* = 0.02). No significant differences emerged between the two groups regarding the other features of the metabolic syndrome (arterial hypertension, obesity or dyslipidemia), alimentary habits (coffee or soft drinks consumption) or leisure physical activity. Alcohol consumers showed higher serum ferritin and urate levels (*p* = 0.02 for both). No differences were detected in non-invasive markers of fibrosis (NFS, FIB-4) or steatosis (HIS) and similarly, two-dimensional transient elastography was comparable between the two groups. Among ultrasound parameters, subcutaneous fat tissue was slightly thicker in abstainers (21 mm vs. 19 mm, *p* = 0.02) with no differences in visceral fat. Cirrhosis was detected slightly more often in alcohol abstainers (28.0% vs. 17.4%, *p* = 0.05) at variance from HCC whose frequency was similar in the two groups. To summarize, a low alcohol intake did not appear to be detrimental in general. However, despite all considered subthreshold for determining cirrhosis as single etiologic agent, the group of alcohol consumers was very heterogeneous. Therefore, alcohol consumers were divided into 4 groups (namely Q1, Q2, Q3, Q4) using quartile range according to lifetime exposure to alcohol as determined by LACU. Patients included in the quartiles from 1 to 3 displayed highly homogeneous characteristics concerning demography, laboratory and clinical features, while Q4 showed different values. Therefore, we decided to aggregate the first three quartiles into a unique group named Q1–Q3 and to consider them altogether. Characteristics of the single quartiles are available in Appendix A.

Patients in Q4 were older than lifetime abstainers and Q1–Q3 (Table 3). Male gender was more frequently represented in any drinking group than in abstainers and in Q4 compared to Q1–Q3. No differences among abstainers and any drinking group were detected as for BMI, leisure activity and alimentary habits. Arterial hypertension was more frequently detected in Q4 compared to Q1–Q3, diabetes mellitus was less represented in Q1–Q3 compared to abstainers. Dyslipidemia was equally represented in the subgroups and no differences were detected in HOMA index in non-diabetic patients. Current alcohol consumption was higher in Q4 compared to Q1–Q3 (*p* < 0.001). Exclusive wine and spirits consumption was equally represented in the two drinkers groups whereas beer (both exclusive and overall) was consumed more frequently in Q1–3 and overall wine and spirits were consumed more frequently in Q4. A binge-drinking pattern was reported more frequently in Q4 (*p* = 0.002). Despite an overall comparable exposure to cigarette smoke, smoking subjects in Q4 reported a slightly higher consumption, as estimated by pack-years calculation, compared to Q1–Q3 = (*p* 0.02). As for laboratory parameters, no differences emerged in glycosylated haemoglobin, serum lipid levels, ALT, AST and gammaGT, urea and serum electrolytes levels. Ferritin was higher in Q4 compared to other subgroups and urate levels were lower in Q4 compared to Q1–3. HIS was comparable among all subgroups. Among ultrasound parameters, subcutaneous adipose tissue was thicker in abstainers compared to Q4 (*p* = 0.007), with no significant differences in visceral adipose tissue. We then assessed the relationship between different degrees of alcohol exposure and markers of fibrosis. All fibrosis diagnostic markers consistently showed the lowest values in Q1–3. In particular, NFS, FIB4 and proportion of patients with clinically evident cirrhosis were lower in Q1–Q3 compared to both abstainers and Q4 (both NFS and cirrhosis clinical evaluation *p* = 0.003 vs. abstainers, *p* < 0.001 vs. Q4, FIB4 *p* = 0.01 vs. abstainers, *p* < 0.001 vs. Q4). 2D-SWE confirmed lower values in Q1–Q3 compared to Q4 (*p* < 0.001) but not to abstainers that showed also lower values than Q4 (*p* = 0.003). Indeed, also HCC prevalence significantly differed between abstainers and LACU subgroups. Q4 displayed the highest prevalence of the disease compared to all the other categories (*p* = 0.006 vs. abstainers, *p* < 0.001 vs. Q1–Q3) but abstainers had higher HCC frequency than Q1–Q3 (*p* < 0.001).

Figure 2 shows the frequency of advanced liver disease in our cohort of patients according to LACU, showing lower prevalence in Q1–Q3.

We then analysed separately the binge drinkers subgroup, that included 35 patients. Compared to non-binge drinkers, binge drinkers were older (median age 62 vs. 58 years, *p* = 0.03), more frequently males (88.2% vs. 82.9%, *p* = 0.04), had drunk alcohol for a longer period of time (median 30 vs. 20 years, *p* = 0.01) and in a greater amount (median current consumption 7.0 vs. 2.5 units/week and median LACU 31.4 vs. 8.6, *p* < 0.001 for both). Overall wine and beer, and exclusive beer and spirits consume were equally represented in binge and non-binge cohorts, whereas wine was exclusively consumed less frequently by binge compared to non-binge drinkers (20% vs. 39.8%, *p* = 0.03) and spirits were overall consumed more frequently by binge compared to non-binge drinkers (54.3% vs. 25.3%, *p* < 0.001). Anthropometric features, co-morbidities, lifestyle habits, laboratory parameters and non-invasive fibrosis markers (NFS, FIB-4 and 2D-SWE) were comparable between the two groups. No differences were noticed in the frequency of cirrhosis (20.0% vs. 16.9%, *p* = n.s.) and HCC (17.1% vs. 9.0%, *p* = n.s.).

Complete data of separated binge and non-binge drinkers are reported in Appendix A.

Focusing on the small ex-drinkers subgroup (17 patients, reporting a median LACU of 7.14 (range 1.71–160.71), we found that they were similar to current drinkers, as for age, sex, smoking exposure, BMI, arterial hypertension, diabetes, dyslipidemia and alimentary habits including alcohol consumption (roughly one third exclusive wine drinkers, 12% binge-drinking) but they reported a slightly reduced physical activity compared to active drinkers (*p* = 0.04). As for laboratory parameters, HDL cholesterol was higher in ex drinkers compared to current drinkers (55 mg/dL vs. 47 mg/dL, *p* = 0.004), while ferritin levels were in between the two groups (95 ng/mL), with no significant differences. Cirrhosis was detected in 5 subjects (29.4%), at a frequency comparable to never and moderate drinkers but slightly higher than in low consumers (*p* = 0.03). HCC was detected in 1 patient (5.9%), with no differences with other subgroups.

We then considered patients according to the kind of alcoholic beverages consumed. Wine was consumed by a total of 165 patients (82.1%), mostly in association with spirits and/or beer (92 subjects, 45.8%). To better evaluate the role of modest amount of wine in our patients with NAFLD, we considered patients reporting drinking exclusively wine and compared them to abstainers and patients not drinking wine; as laboratory parameters did not differ significantly between groups, these data are not reported (Table 4).

Wine was consumed exclusively by 73 patients (36.3%). Compared to non-wine drinkers (36 subjects: 28 exclusively beer-drinkers, 7 exclusively spirits drinkers and one consumer of beer plus spirits), wine consumers were similar for sex, BMI distribution, frequency of arterial hypertension and diabetes, though they were older (median age 65 vs. 40 years, *p* < 0.001), less physically active (1.19 vs. 1.52, *p* = 0.02), more frequently dyslipidemic (74% vs. 42%, *p* < 0.001) and with a longer period of alcohol consumption (median drinking years 30 vs. 15, *p* < 0.001, median LACU 25.0 vs. 4.3, *p* = 0.003). Wine exclusive drinkers displayed higher values of non-invasive fibrosis markers (median NFS −0.91 vs. −2.37, *p* = 0.001, median FIB-4 1.32 vs. 0.73, *p* = 0.009, median 2D-SWE 6.1 kPa vs. 5.7 kPa, *p* = 0.04). Overall cirrhosis was detected in 13 exclusive wine drinkers (17.8%) and in 2 non-wine drinkers (5.6%), *p* not significant; no HCC were reported in non-wine drinkers whereas 8 cases were detected in exclusive wine drinkers (11.0%).

Beer was consumed exclusively by 28 patients (13.9%). Compared non-beer drinkers (108 subjects: 73 exclusively wine drinkers, 7 exclusively spirits drinkers and 28 consumers of wine plus spirits), beer drinkers were younger (median age 41 vs. 65, *p* < 0.001), with no differences in sex distribution; despite a similar BMI, they exerted a more vigorous physical activity (mean 1.53 vs. 1.17, *p* = 0.04) and were less frequently dyslipidemic (11 patients, 39.3% vs. 68 patients, 63.0%, *p* = 0.02) with comparable rates of arterial hypertension and diabetes. Beer drinkers reported a lower exposition to alcoholic beverages, both currently (median 1 vs. 4 units per week, *p* = 0.01) and lifelong (median drinking years 13 vs. 30, *p* < 0.001, median LACU 4.3 vs. 30.0, *p* 0.001) and displayed a less advanced liver disease (median NFS −2.29 vs. −0.83, *p* = 0.001, median FIB-4 0.75 vs. 1.53, *p* = 0.01, median 2D-SWE 5.7 kPa vs. 6.7 kPa, *p* = 0.02; no cirrhosis or HCC were noticed in exclusive beer drinkers at variance with non-beer consumers where 21 cases of cirrhosis and 15 cases of HCC were detected.

Spirits were consumed exclusively by only 7 patients (3.5% of the whole drinkers cohort, 86% males, median age 45); no differences were detected in all the parameters evaluated between exclusive spirits and non-spirits consumers (142 patients). Among exclusive spirits consumers, cirrhosis was detected in 2 patients (28.6%) without any HCC case.

### 3.3. Association between Patient Characteristics and Advanced Liver Disease

#### 3.3.1. Cirrhosis

Factors associated with advanced fibrosis included higher age, greater waist circumference and BMI, higher levels of HbA1c, gammaGT and AST, LACU (but not current alcohol consumption), smoking exposure as quantified by pack-years, history of arterial hypertension and diabetes mellitus (Table 5). More intense leisure physical activity and beer consumption were found to be associated to less advanced fibrosis. Cirrhotic patients were found to have lower serum levels of cholesterol, albumin and platelet, which should be held as manifestations of cirrhosis rather than risk factors.

On multivariate logistic regression analysis (Table 6), presence of cirrhosis was confirmed to be positively associated to older age and a history of hypertension and diabetes mellitus while very low current (C1) or lifelong (LACU Q1–Q3) alcohol intake appeared to be protective compared to complete abstinence. At variance, a higher alcohol consumption, but always in the range of the NAFLD definition, was found to be possibly harmful when protracted lifelong.

#### 3.3.2. Hepatocellular Carcinoma (HCC)

Overall, 32 HCC were detected in our cohort of patients, in 10 cases non associated with cirrhosis. Factors associated with HCC included older age, male gender, history of arterial hypertension and diabetes mellitus, alcohol consumption as evaluated by current alcohol use, LACU and drinking years, overall spirits consumption and presence of cirrhosis. More intense leisure physical activity and beer consumption were found to be associated to lower HCC frequency (Table 7).

HCC patients were found to have lower serum levels of total cholesterol, albumin and platelet and higher levels of gammaGT and AST but these findings probably reflect a higher proportion of patients with advanced liver disease.

On multivariate logistic regression analysis (Table 8), presence of HCC was confirmed to be positively associated with older age, male sex, history of cirrhosis and arterial hypertension (all known risk factors for HCC), whereas a very low current (C1) and lifetime (Q1–3) alcohol intake was associated with a reduced frequency of HCC compared to complete abstinence.

## 4. Discussion

Fatty liver is the hepatic manifestation of a systemic metabolic dysfunction, which is heterogeneous in its underlying conditions, presentation, course and outcomes. Considering its direct hepato-toxic effect, moderate alcohol consumption has been considered detrimental in patients with underlying chronic liver diseases of different origin, including NAFLD, for a long time, regardless of any level of alcohol intake. In the last decades a great amount of data reporting a J-shaped relationship between alcohol consumption and extra-hepatic events (as all cause mortality and cardiovascular disease) has been reported, with a moderate alcohol consumption (1–2 units per day) associated with better outcomes compared to abstainers [19,20], even in NAFLD patients [21]. Similarly, data of a possible beneficial effect of very modest amount of alcohol on the course of NAFLD are accumulating, creating a hot controversy about alcohol consumption in these patients [10,11].

In this study, we examined 276 patients with NAFLD at their first visit in our tertiary hepatology centre. We evaluated the association between lifestyle, alcohol consumption and metabolic syndrome features and the severity of liver disease, namely cirrhosis and hepatocellular carcinoma. For a more accurate evaluation of overall alcohol exposure, besides current alcohol consumption (expressed as units per week), that could be influenced by recent conditions including the awareness of liver disease, we designed a new parameter, that we called LACU (Lifetime Alcohol Consumption Units). One LACU was defined as 7 alcoholic units (70 g of pure alcohol) per week for one year. We chose this parameter paralleling the pack/years used to estimate the toxic exposure to cigarette smoking and we decided to use weekly instead of daily amount of alcohol to correctly evaluate social drinkers that preferably consume alcohol during weekends.

In our cohort of patients fulfilling NAFLD criteria (daily ethanol consumption < 210 g in men and <140 g in women and exclusion of other known causes of liver disease), a very low alcohol use, far below the subthreshold amount, as either current or lifelong exposure, was associated with a lower frequency of cirrhosis at univariate and multivariate analysis, compared to both abstainers and moderate consumers (corresponding to C2 and Q4).

In multivariate analysis, a low lifetime alcohol exposure (LACU) was also independently associated with a reduced prevalence of HCC compared to both abstainers and moderate consumers. This was evident for lifetime but not for current consume, suggesting a possible protective effect for a chronic very low exposure sustained over time. Overall, these findings may appear counterintuitive, as we would have expected, starting from abstainers, a progressive increase in risk with progressive increase in alcohol exposure. Conversely, when looking at NAFLD patients consuming an amount of alcohol below the most utilized thresholds associated with development of liver disease, the behaviour did not appear progressive but rather diverging: the group of patients with a greater intake showed the expected detrimental consequences of alcohol, whereas patients with very low alcohol intake showed better outcomes than abstainers, which is rather an innovative finding for liver diseases. Our cohort did not allow relevant statistical evaluations on specific effects of different kinds of alcoholic beverages or drinking patterns being composed by prevalently regularly wine-drinkers patients (wine drinking reported by 82% of our patients, in an exclusive fashion in roughly one third of subjects, and with a non-binge pattern of consumption in more than 80% of cases).

Indeed, though still debated, many previous studies have suggested a positive role of low amount of alcohol, especially wine, on NAFLD [13,14,22,23,24], through beneficial effects on insulin sensitivity, plasma adiponectin and systemic inflammation [10,25,26,27], but a potential protective effect on hepatocellular carcinoma was never convincingly demonstrated. Studies reporting a detrimental effect of alcohol use on hepatocellular carcinoma in NAFLD patients are highly heterogeneous, as some focused on patients with an already advanced liver disease [7] and others considered altogether patients inside NAFLD criteria without differentiation between low and moderate alcohol consume [9,28]. A recent review of six longitudinal observational cohort studies concluded that any level of alcohol intake in NAFLD may be harmful to liver health, but the studies are highly heterogeneous in definition of quantitative and qualitative alcohol exposure, lifetime consumption was not extensively evaluated, and no separate analysis was conducted on very low alcohol exposure [29]. At variance, a very low alcohol consumption (less than two times per week and/or <10 g ethanol per day, preferably wine) was associated with a lower risk of advanced liver disease including hepatocellular carcinoma in sub-analysis of prospective studies on large Finnish and Korean populations [8,21].

Hepatocellular carcinoma can develop in patients with fatty liver even in pre-cirrhotic stage [30], being the strongest modifiable risk factors represented by diabetes, arterial hypertension [31,32] and cigarette smoking [33]. The beneficial effect on insulin-resistance, the pivotal feature of metabolic syndrome, and as a direct consequence on the development of diabetes, may constitute the key point of the protective effect of very low amount of alcohol on hepatocarcinogenesis, at variance with data on cancers of different origin, especially breast, that are associated with any alcohol consumption [11,21]. This effect may be more evident in our cohort of Mediterranean patients, more overweight than obese (median BMI 29.0) and preferable wine consumers substantially stable over time. Wine consumption has been widely associated with a healthier, Mediterranean diet, rich of vegetables, and a more active lifestyle [34]. In addition, non-alcoholic constituents of wine, as quercetin and resveratrol, have been shown to reduce oxidative damage and inhibit activation of hepatic stellate cells both in in vitro and in vivo experimental settings [35,36,37]. We can speculate that these characteristics of light wine-drinkers may contribute to a more favourable microbiota profile leading to a lower grade of intestinal end systemic inflammation that is known to be associated to disease progression and HCC in NAFLD patients [38,39].

Limitations of the present study are the relatively small patient population (which allowed however more accurate characterisation of lifestyle habits compared to large anonymous databases), especially of binge and non-wine drinkers, that does not allow to draw reliable conclusions on the effects of specific kind or patterns of alcoholic drinking, and the observational nature which promotes observation of associations but cannot be conclusive on causative effects.

## 5. Conclusions

In conclusion, our data support a slightly more permissive approach to selected regular drinkers of very low amounts of alcohol: in NAFLD patients without significant fibrosis, an overall healthier lifestyle based on Mediterranean diet including up to one glass of wine daily, may not be harmful and instead may possibly reduce the risk of progression of liver disease, though the mechanism behind this effect remains to be elucidated.

## Figures and Tables

**Figure 1 nutrients-14-02493-f001:**
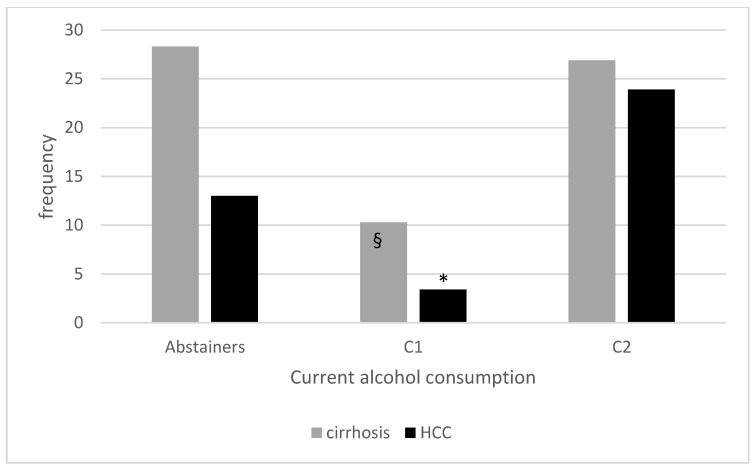
Prevalence of cirrhosis and HCC according to current alcohol consumption. § *p* < 0.001 compared to current abstainers and <0.01 compared to C2; * *p* < 0.01 compared to current abstainers and < 0.001 compared to C2.

**Figure 2 nutrients-14-02493-f002:**
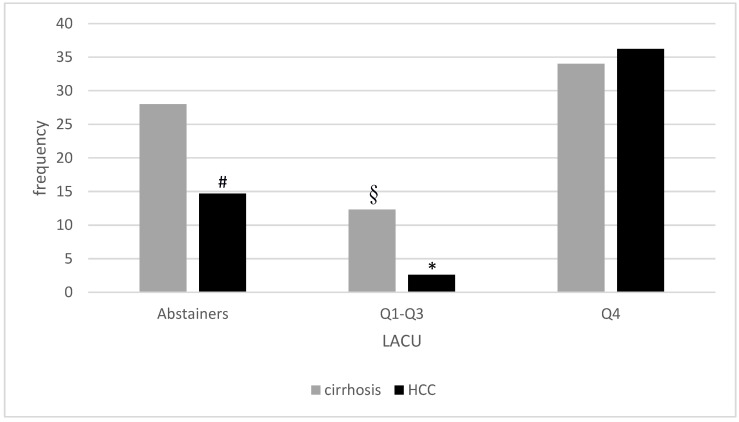
Prevalence of cirrhosis and HCC according to lifetime alcohol consumption (LACU). § *p* < 0.01 compared to lifetime abstainers and *p* < 0.001 compared to Q4; * *p* < 0.001 compared to both lifetime abstainers and Q4; # *p* < 0.01 compared to Q4.

**Table 1 nutrients-14-02493-t001:** Clinical, demographic, laboratory and ultrasound characteristics of all patients with NAFLD (*n* = 276).

Variable	All Patients *n* = 276
Age at first visit (years)	59 (18–88)
Male sex	162 (58.7%)
Waist (cm)	104 (60–160)
BMI (kg/m^2^)	29.0 (18.6–48.8)
Hypertension	158 (57.2%)
DM type 2	74 (26.8%)
Dyslipidemia	178 (64.4%)
Smoking habit, ever	128 (46.4%)
Among smokers, pack-years (p/years)	18.8 (0.5–100)
Leisure physical activity	1.25 ± 0.71
Coffee cups per day	2 (0–9)
Soft drinks servings per week	2 (0–7)
HbA1c (mmol/moL)	42 (23–109)
Total cholesterol (mg/dL)	196 (69–353)
HDL cholesterol (mg/dL)	48 (11–119)
LDL cholesterol (mg/dL)	122 (51–231)
Triglycerids (mg/dL)	119 (35–934)
Ferritin (ng/mL)	98 (7–2098)
Urate (mg/dL)	5.8 (2.5–10.2)
urea (mg/dL)	35 (11–101)
AST (U/L)	30 (11–200)
ALT (U/L)	34 (8–275)
Total bilirubin (mg/dL)	0.7 (0.3–3.0)
Platelet count (×10^3^/μL)	223 (27–402)
gammaGT (U/L)	51 (8–905)
Albumin (g/dL)	4.1 (2.9–4.9)
HOMA (DM excluded)	2.3 (0.8–11.6)
NFS	−1.10 (−4.57/+3.98)
FIB-4	1.27 (0.03–8.50)
HSI	40.6 (30.0–61.6)
Subcutaneous fat (mm)	20 (10–85)
Visceral fat (mm)	69 (21–126)
2D-SWE (kPa)	6.5 (3.3–50.5)
Cirrhosis	56 (20.3%)
HCC	32 (11.6%)

Data are expressed as median (range) or absolute number (%) except for leisure physical activity which is expressed as mean ± standard deviation. Leisure-time physical activity was classified as 1 (up to 30 min per week), 2 (30–90 min) and 3 (more than 90 min). BMI (body mass index), DM (diabetes mellitus), NFS (NAFLD fibrosis score), FIB-4 (fibrosis-4), HIS (hepatic steatosis index), 2D-SWE (two dimensional shear-wave elastography), HCC (hepatocellular carcinoma).

**Table 2 nutrients-14-02493-t002:** Patient characteristics according to current alcohol intake per week.

Variable	Current Abstainers *n* = 92	Current Drinkers *n* = 184	*p*	C1 *n* = 117	C2 *n* = 67	*p*
Age (years)	61 (18–85)	58 (18–88)		54 (18–78)	65 (18–88)	*a, c*
Male sex	35 (37.6%)	127 (69.0%)	*<0.001*	79 (67.5%)	48 (71.6%)	*a, b*
Waist (cm)	104 (76–160)	104 (60–142)		103 (75–124)	106 (60–142)	
BMI	29.2 (20.1–44.0)	28.9(18.6–48.8)		29.1(20.8–42.2)	27.9(18.6–48.8)	
Hypertension	51 (54.8%)	107 (58%)		63 (53.8%)	44 (65.7%)	
DM type 2	30 (32.6%)	44 (23.9%)	*0.03*	23 (19.7%)	21 (31.3%)	
Dyslipidemia	62 (67.4%)	116 (63.0%)		71 (60.7%)	45 (67.2%)	
Smoking habit, ever	35 (38.0%)	93 (50.5%)	*0.06*	54 (46.2%)	39 (58.2%)	
Among smokers (p/years)	15.6 (0.5–100)	20 (0.5–100)		15 (1–90)	30 (0.5–100)	
Current alcohol units/week	0	3 (1–21)	*<0.001*	2 (1–6)	14 (7–21)	*c*
LACU	0 (0–160.7)	12.9 (0.7–200.0)		6.7 (0.7–122.9)	50 (3.0–200.0)	*b, c*
Drinking years	0 (0–60)	25 (1–60)		20 (1–55)	30 (3–60)	*a, b, c*
Binge-drinkers	3 (3.2%)	32 (17.4%)	*<0.001*	14 (12.0%)	18 (26.9%)	*a, b, c*
Exclusive wine drinkers	5 (5.4%)	68 (37.0%)	*<0.001*	41 (35.0%)	27 (40.3%)	*a, b*
Overall wine drinkers	14 (15.2%)	151 (82.1%)	*<0.001*	87 (74.4%)	64 (95.5%)	*a, b, c*
Exclusive beer drinkers	3 (3.2%)	25 (13.6%)	*0.007*	23 (19.7%)	2 (3.0%)	*a, c*
Overall beer drinkers	11 (12.0%)	82 (44.5%)	*<0.001*	61 (52.1%)	21 (31.3%)	*c*
Exclusive spirits drinkers	0	7 (3.8%)		6 (5.1%)	1 (1.6%)	
Overall spirits drinkers	9 (9.8%)	50 (27.2%)	*<0.001*	21 (17.9%)	29 (43.3%)	*c*
Leisure physical activity	1.14 ± 0.76	1.30 ± 0.68		1.35 ± 0.69	1.20 ± 0.68	
Coffee cups per day	2 (0–6)	2 (0–9)	*0.04*	2 (0–9)	2 (0–7)	
Soft drinks per week	2 (0–6)	2 (0–7)		2 (0–6)	2 (0–7)	
HbA1c (mmol/moL)	48 (23–109)	41 (25–94)	*0.05*	40 (34–94)	41 (25–103)	
Total cholesterol (mg/dL)	197 (69–353)	194 (84–328)		192 (120–256)	196 (84–309)	
HDL cholesterol (mg/dL)	50 (29–119)	47 (11–86)	*0.04*	46 (29–86)	48 (11–84)	
LDL cholesterol (mg/dL)	129 (55–231)	121 (50–208)		119 (55–184)	131 (50–201)	
Triglycerids (mg/dL)	114 (35–934)	124 (32–810)		122 (44–810)	126 (32–667)	
Ferritin (ng/mL)	66 (7–876)	140 (7–2098)	*0.03*	91 (7–349)	162 (7–2098)	
Urate (mg/dL)	5.7 (2.5–7.7)	6.0 (2.5–10.2)		6.1 (2.5–10.2)	5.8 (3.8–6.6)	
urea (mg/dL)	35 (11–52)	37 (21–101)		36 (21–40)	37 (21–101)	
AST (U/L)	32 (13–200)	30 (11–200)		30 (18–200)	27 (11–80)	
ALT (U/L)	37 (8–230)	34 (11–275)		38 (15–275)	30 (11–152)	
Total bilirubin (mg/dL)	0.7 (0.3–2.8)	0.7 (0.3–3.0)		0.7 (0.3–1.6)	0.7 (0.3–3.0)	
Platelet count (×10^3^/μL)	223 (48–400)	225 (44–402)		237 (44–402)	212 (81–393)	
gammaGT (U/L)	52 (8–905)	51 (15–410)		53 (18–410)	48 (15–406)	
Albumin (g/dL)	4.1 (2.9–5.0)	4.1 (2.6–4.9)		4.2 (3.2–4.7)	4.1 (2.6–4.9)	
HOMA (DM excluded)	2.2 (1.4–8.9)	2.3 (0.5–11.6)		1.8 (1.4–5.2)	3.8 (0.5–11.6)	*c*
NFS	−1.2 (−3.6/+2.8)	−1.2 (−4.6/+4.0)		−1.9(−4.6/+4.0)	−0.7(−4.4/+2.6)	*c*
FIB-4	1.35 (0.31–12.74)	1.21 (0.19–8.5)		1.00 (0.23–8.5)	1.40 (0.19–6.3)	
HSI	40.9 (30.1–57.8)	40.3(30.0–61.6)		40.8 (31.8–61.6)	39.4 (30.0–56.6)	
Subcutaneous fat (mm)	20 (10–60)	19 (10–85)		20 (10–47)	19 (10–85)	
Visceral fat (mm)	68 (34–126)	70 (12–179)		67 (15–120)	72 (12–179)	
2D-SWE (kPa)	6.8 (3.7–27.2)	6.5 (3.3–30.2)		6.0 (3.3–22.1)	7.8 (3.5–50.1)	*a, c*
Cirrhosis	26 (28.3%)	30 (16.3%)	*0.02*	12 (10.3%)	18 (26.9%)	*a, c*
HCC	12 (13.0%)	20 (10.9%)		4 (3.4%)	16 (23.9%)	*a, c*

Current abstainers include 17 ex-drinkers. Active drinkers were divided in C1 (very low consumers, < 70 g per week), C2 (moderate consumers, 70–210 g per week for men and 70–140 g per week for women). Data are expressed as median (range) or absolute number (%) except for leisure physical activity which is expressed as mean ± standard deviation. Only statistically significant differences are reported in the p columns. In p column, *a* means *p* < 0.0167 between current abstainers and C1, *b* means *p* < 0.0167 between current abstainers and C2, *c* means *p* < 0.0167 between C1 and C2.

**Table 3 nutrients-14-02493-t003:** Patient characteristics according to lifetime alcohol exposure (LACU). LACU aggregate.

Variable	Abstainers *n* = 75	Consumers 201	*p*	Q1–Q3 *n* = 154	Q4 *n* = 47	*p*
Age (years)	59 (18–80)	58 (18–88)		56 (18–78)	71 (45–86)	*b, c*
Male sex	25 (33.3%)	137 (68.2%)	*<0.001*	101 (65.6%)	36 (76.6%)	*a, b*
Waist (cm)	103 (88–160)	104 (60–142)		104 (60–142)	109 (85–142)	
BMI	28.8 (24.2–44.0)	29.1 (18.6–48.8)		29.1 (18.6–48.8)	28.4 (23.2–40.0)	
Hypertension	43 (56.7%)	115 (57.2%)		81 (52.6%)	34 (72.3%)	*c*
DM type 2	26 (35.7%)	48 (23.9%)	*0.03*	33 (21.4%)	15 (31.9%)	*a*
Dyslipidemia	52 (69.3%)	126 (62.7%)		95 (61.7%)	31 (66.0%)	
Smoking habit, ever	28 (36.8%)	100 (49.8%)	*0.04*	76 (49.4%)	24 (51.1%)	
Among smokers p/years	20.0 (0.5–90)	17.9 (0.5–100)		15.0 (0.5–100)	30.0 (3–100)	
Current alcohol units/week	0	3 (0.5–21.0)	*<0.001*	2 (0–21)	14 (0–21)	*c*
LACU	0	12.9 (0.7–200.0)	*<0.001*	8.6 (0.7–40.0)	80.0 (42.8–200.0)	*c*
Drinking years	0	24 (1–60)	*<0.001*	20 (1–60)	40 (5–60)	*a, b, c*
Binge-drinkers	0	35 (17.4%)		18 (11.7%)	17 (36.2%)	*c*
Exclusive wine drinkers	0	73 (36.3%)		55 (35.7%)	18 (38.3%)	
Overall wine drinkers	0	165 (82.1%)		120 (77.9%)	45 (95.7%)	*c*
Exclusive beer drinkers	0	28 (13.9%)		28 (18.2%)	0	
Overall beer drinkers	0	93 (46.3%)		79 (51.3%)	14 (29.8%)	*c*
Exclusive spirits drinkers	0	7 (3.5%)		5 (3.2%)	2 (4.1%)	
Overall spirits drinkers	0	59 (29.4%)		27 (13.4%)	32 (68%)	*c*
Leisure physical activity	1.20 ± 0.80	1.27 ± 0.68		1.29 ± 0.72	1.17 ± 0.71	
Coffee cups per day	2 (0–6)	2 (0–9)		2 (0–9)	2 (0–4)	
Soft drinks per week	2 (0–6)	2 (0–7)		2 (0–7)	2 (0–6)	
HbA1c (mmol/mol)	49 (23–109)	41 (27–103)	*0.02*	41 (27–103)	38 (28–85)	
Total cholesterol (mg/dL)	196 (69–353)	195 (84–328)		195 (84–309)	195 (99–328)	
HDL cholesterol (mg/dL)	49 (29–76)	48 (11–119)		47 (12–86)	51 (11–119)	
LDL cholesterol (mg/dL)	128 (55–231)	119 (32–208)		119 (32–201)	126 (51–208)	
Triglycerids (mg/dL)	114 (42–934)	123 (32–810)		122 (32–810)	124 (49–287)	
Ferritin (ng/mL)	65 (7–876)	115 (7–2098)	*0.02*	87 (7–761)	244 (20–2098)	*b, c*
Urate (mg/dL)	5.7 (2.5–7.7)	5.9 (2.5–10.2)	*0.02*	6.3 (2.5–10.2)	5.2 (3.8–7.4)	*c*
urea (mg/dL)	35 (11–52)	36 (21–101)		33 (21–101)	38 (21–49)	
AST (U/L)	32 (13–200)	30 (11–200)		30 (14–200)	28 (11–92)	
ALT (U/L)	37 (8–230)	34 (13–275)		38 (13–275)	29 (15–151)	
Total bilirubin (mg/dL)	0.67 (0.3–2.8)	0.71 (0.4–3.0)		0.70 (0.3–3.0)	0.76 (0.4–2.3)	
Platelet count (×10^3^/µL)	224 (48–400)	223 (44–402)		241 (44–402)	202 (44–298)	*c*
gammaGT (U/L)	52 (8–905)	51 (16–410)		48 (12–410)	57 (18–301)	
Albumin (g/dL)	4.1 (2.9–4.8)	4.1 (2.6–4.9)		4.2 (2.6–4.9)	4.0 (2.9–4.7)	*c*
HOMA (DM excluded)	1.9 (1.4–8.9)	2.4 (0.5–11.6)		2.3 (0.5–11.6)	3.8 (1.8–6.2)	
NFS	−1.5 (−3.6/+1.1)	−1.1 (−4.6/+4.0)		−1.8 (−4.6/+4.0)	0.2 (−2.7/+2.8)	*a, c*
FIB-4	1.28 (0.3–12.7)	1.21 (0.2–8.5)		0.99 (0.2–8.5)	1.8 (0.7–6.3)	*a, c*
HSI	40.4 (30.0–53.8)	40.6 (30.1–61.6)		40.6 (30.1–61.6)	40.6 (33.0–50.6)	
Subcutaneous fat (mm)	21 (15–60)	19 (10–85)	*0.02*	19 (10–85)	20 (10–27)	*b*
Visceral fat (mm)	67 (35–110)	71 (21–126)		71 (21–126)	72 (36–116)	
2D-SWE (kPa)	6.8 (3.7–27.2)	6.5 (3.3–50.5)		6.0 (3.3–50.5)	12.0 (5.5–40.8)	*b, c*
Cirrhosis	21 (27.6%)	35 (17.4%)	*0.05*	19 (12.3%)	16 (34.0%)	*a, c*
HCC	11 (14.5%)	21 (10.4%)		4 (2.6%)	17 (36.2%)	*a, b, c*

Q1–3 (<4.29–40.00 LACU), Q4 (>40.01 LACU). Data are expressed as median (range) or absolute number (%) except for leisure physical activity which is expressed as mean ± standard deviation. Only statistically significant differences are reported in the *p* columns. In *p* column, *a* means *p* < 0.0167 between abstainers and Q1–Q3, *b* means *p* < 0.0167 between abstainers and Q4, *c* means *p* < 0.0167 between Q1–3 and Q4.

**Table 4 nutrients-14-02493-t004:** Patients characteristics according to the kind of alcoholic beverages consumed.

Variable	Abstainers *n* = 75	Exclusive Wine Drinkers *n* = 73	Non Wine Drinkers *n* = 36	*p*
Age (years)	59 (18–80)	65 (37–88)	40 (18–67)	*a, b, c*
Male sex	25 (33.3%)	40 (54.8%)	27 (75%)	*b*
BMI	28.8 (24.2–44.0)	29.6 (20.8–40.3)	28.6 (18.6–37.1)	
Hypertension	43 (56.7%)	51 (69.9%)	18 (50%)	
DM type 2	26 (35.7%)	18 (24.6%)	6 (16.7%)	
Dyslipidemia	52 (69.3%)	54 (74.0%)	15 (41.7%)	*b, c*
Smoking habit, ever	28 (36.8%)	37 (50.7%)	17 (47.2%)	
Among smokers p/years	20.0 (0.5–90)	26 (5–50)	9 (3–50)	
Current alcohol units/week	0	3 (0.0–21.0)	1 (0.0–21.0)	*a, b*
LACU	0	25.0 (2.9–150.0)	4.3 (0.7–90.0)	*a, b, c*
Drinking years	0	30 (10–60)	15 (1–30)	*a, b, c*
Binge-drinkers	0	10 (13.7%)	6 (16.7%)	
Leisure physical activity	1.20 ± 0.80	1.19 ± 0.68	1.52 ± 0.83	*c*
Coffee cups per day	2 (0–6)	2 (0–6)	2 (0–5)	
Soft drinks per week	2 (0–6)	2 (0–5)	2 (0–7)	
NFS	−1.5 (−3.6/+1.1)	−0.9 (−3.3/+2.0)	−2.4 (−4.6/+1.1)	*c*
FIB-4	1.28 (0.3–12.7)	1.32 (0.2–4.3)	0.73 (0.3–2.6)	*c*
Visceral fat (mm)	67 (35–110)	74 (21–126)	75 (32–120)	
2D-SWE (kPa)	6.8 (3.7–27.2)	6.1 (4.0–28.8)	5.7 (3.3–20.0)	*b*
Cirrhosis	21 (27.6%)	13 (17.8%)	2 (5.6%)	*b*
HCC	11 (14.5%)	8 (11.0%)	0 (0%)	

Data are expressed as median (range) or absolute number (%). Only statistically significant differences are reported in the p column. *a: p* < 0.0167 between abstainers and exclusive wine drinkers; *b: p* < 0.0167 between abstainers and non-wine drinkers; *c: p* < 0.0167 between exclusive wine and non-wine drinkers.

**Table 5 nutrients-14-02493-t005:** Patient characteristics according to cirrhosis status (overall clinical evaluation).

Variable	Non Cirrhosis *n* = 220	Cirrhosis *n* = 56	*p*
Age (years)	56 (15–88)	68 (45–86)	*<0.001*
Male sex	129 (58.6%)	33 (58.9%)	
Waist (cm)	103 (75–142)	114 (93–142)	*0.005*
BMI	29.0 (18.6–42.2)	29.4 (20.9–48.8)	*0.03*
Hypertension	112 (50.9%)	46 (87.5%)	*<0.001*
DM type 2	37 (16.8%)	37 (66.1%)	*<0.001*
Dyslipidemia	143 (65.0%)	35 (62.5%)	
Smoking habit, ever	100 (45.4%)	28 (50.0%)	
Among smokers (p/years)	15 (0.5–100)	32.5 (1–90)	*0.03*
Current alcohol units/week	2 (0–21)	1 (0–21)	
LACU	5.9 (0–160)	5.0 (0–200)	*0.002*
Drinking years	20 (0–60)	20 (0–60)	
Binge-drinkers	28 (12.7%)	7 (12.5%)	
Exclusive wine drinkers	56 (25.4%)	17 (30.3%)	
Overall wine drinkers	132 (60%)	33 (58.9%)	
Exclusive beer drinkers	28 (12.7%)	0 (0%)	
Overall beer drinkers	85 (38.6%)	8 (14.3%)	*<0.001*
Exclusive spirits drinkers	5 (2.2%)	2 (3.6%)	
Overall spirits drinkers	44 (20%)	15 (26.8%)	
Leisure physical activity	1.33 ± 0.69	0.77 ± 0.65	*<0.001*
Coffee cups per day	2 (0–9)	2 (0–4)	
Soft drinks per week	2 (0–6)	2 (0–6)	
HbA1c (mmol/mol)	42 (23–98)	51 (25–109)	*0.005*
Total cholesterol (mg/dL)	200 (69–353)	176 (98–327)	*<0.001*
HDL cholesterol (mg/dL)	48 (12–119)	43 (11–76)	*0.02*
LDL cholesterol (mg/dL)	125 (32–212)	109 (51–231)	
Triglycerids (mg/dL)	123 (19–934)	110 (44–278)	
Ferritin (ng/mL)	98 (7–761)	59 (7–2098)	
Urate (mg/dL)	6.0 (3.4–10.2)	5.6 (2.5–7.4)	
urea (mg/dL)	34 (21–101)	36 (11–52)	
AST (U/L)	29 (11–200)	38 (16–124)	*0.004*
ALT (U/L)	34 (8–275)	36 (11–150)	
Total bilirubin (mg/dL)	0.70 (0.26–3.02)	0.76 (0.30–2.32)	
Platelet count (×10^3^/μL)	236 (111–402)	145 (44–368)	*<0.001*
gammaGT (U/L)	38 (8–306)	120 (18–905)	*<0.001*
Albumin (g/dL)	4.2 (2.6–5.0)	3.9 (2.9–4.6)	*<0.001*
HOMA (DM excluded)	2.2 (0.5–11.6)	3.2 (2.7–3.8)	
NFS	−1.6 (−4.6/+1.8)	1.5 (−2.7/+4.9)	*<0.001*
FIB-4	1.01 (0.03–3.47)	3.22 (0.61–12.7)	*<0.001*
HSI	40.6 (30.0–61.6)	40.6 (30.1–54.8)	
Subcutaneous fat (mm)	20 (10–85)	17 (10–60)	
Visceral fat (mm)	67 (12–123)	90 (33–179)	*<0.001*
2D-SWE (kPa)	5.9 (3.3–12)	20 (11.1–56.4)	*<0.001*
HCC	9 (4.1%)	23 (41.1%)	*<0.001*

Data are expressed as median (range) or absolute number (%) except for leisure physical activity which is expressed as mean ± standard deviation. Only statistically significant differences are reported in the *p* column.

**Table 6 nutrients-14-02493-t006:** Multivariate analysis of parameters associated with cirrhosis.

Current Alcohol Consumption				LACU			
**Parameter**	**OR**	**95% CI**	** *p* **	**Parameter**	**OR**	**95% CI**	** *p* **
Male sex	1.32	0.58–3.01	0.498	Male sex	1.03	0.48–2.20	0.946
Age	1.06	1.03–1.10	<0.001				
BMI	1.04	0.95–1.13	0.411	BMI	1.02	0.94–1.10	0.663
Hypertension	2.01	0.86–4.71	0.109	Hypertension	2.69	1.20–6.05	0.017
Diabetes	8.01	3.94–16.29	<0.001	Diabetes	9.33	4.60–18.92	<0.001
**Alcohol intake**				**Alcohol intake**			
Abstinent	Reference			Abstinent	Reference		
C1	0.40	0.18–0.90	0.027	Q1–3	0.42	0.19–0.95	0.037
C2	1.03	0.40–2.60	0.962	Q4	1.35	0.54–3.36	0.010

**Table 7 nutrients-14-02493-t007:** Patient characteristics according to the presence of hepatocellular carcinoma (HCC).

Variable	Non HCC *n* = 244	HCC *n* = 32	*p*
Age (years)	57 (18–88)	72 (52–86)	*<0.001*
Male sex	136 (55.7%)	26 (81.3%)	*0.006*
Waist (cm)	104 (60–160)	106 (70–139)	
BMI	29.1 (18.6–48.8)	28.9 (20.4–44.0)	
Hypertension	129 (52.9%)	29 (90.6%)	*<0.001*
DM type 2	53 (21.7%)	21 (65.6%)	*0.001*
Dyslipidemia	159 (65.1%)	19 (59.4%)	
Smoking habit. ever	110 (45.1%)	18 (56.3%)	
Among smokers (p/years)	15 (1–100)	32 (16–120)	*0.01*
Alcohol units per week	1.8 (0–21)	4.0 (0–21)	*<0.001*
LACU	5.7 (0–200)	43.9 (0–200)	*<0.001*
Drinking years	15 (0–50)	30 (0–60)	*0.002*
Binge-drinkers	31 (12.7%)	4 (12.5%)	
Exclusive wine drinkers	65 (26.6%)	7 (21.9%)	
Overall wine drinkers	145 (59.4%)	20 (62.5%)	
Exclusive beer drinkers	28 (11.5%)	0 (0%)	
Overall beer drinkers	91 (37.3%)	2 6.3%)	*<0.001*
Exclusive spirits drinkers	7 (2.9%)	0 (0%)	
Overall spirits drinkers	48 (19.7%)	11 (34.3%)	*0.056*
Leisure physical activity	1.27 ± 0.69	0.73 ± 0.71	*0.01*
Coffee consumption per day	2 (0–9)	2 (0–3)	
Soft drinks consumption per week	2 (0–6)	2 (0–6)	
HbA1c (mmol/mol)	42 (23–109)	53 (37–103)	
Total cholesterol (mg/dL)	198 (69–353)	177 (99–224)	*0.004*
HDL cholesterol (mg/dL)	48 (11–148)	45 (18–67)	
LDL cholesterol (mg/dL)	124 (12–231)	112 (57–161)	
Triglycerids (mg/dL)	121 (19–934)	114 (44–214)	
Ferritin (ng/mL)	89 (7–2098)	64 (7–402)	
Urate (mg/dL)	5.9 (2.5–10.2)	5.3 (4.0–7.4)	
urea (mg/dL)	35 (11–101)	36 (21–49)	
AST (U/L)	30 (11–124)	37 (13–89)	*0.04*
ALT (U/L)	35 (8–275)	34 (10–151)	
Total bilirubin (mg/dL)	0.70 (0.3–3.0)	0.80 (0.5–2.3)	*0.06*
Platelet count (×10^3^/μL)	228 (44–402)	167 (71–285)	*<0.001*
gammaGT (U/L)	46 (8–905)	100 (18–449)	*<0.001*
Albumin (g/dL)	4.2 (2.6–5.0)	4.0 (2.9–4.6)	*0.002*
HOMA (DM excluded)	3.2 (0.8–11.6)	4.1 (1.6–8.8)	
NFS	−1.5 (−4.6/+4.0)	0.7 (−2.8/+2.8)	*<0.001*
FIB-4	1.13 (0.19–12.74)	2.65 (1.34–6.30)	*<0.001*
HSI	40.6 (30.0–61.6)	40.4 (30.1–53.1)	
Subcutaneous fat (mm)	20 (10–85)	20 (14–31)	
Visceral fat (mm)	69 (55–179)	76 (65–123)	
2D-SWE (kPa)	6.1 (3.3–50.0)	19.8 (6.0–56.4)	*<0.001*
Cirrhosis	33 (13.5%)	23 (71.9%)	*<0.001*

Data are expressed as median (range) or absolute number (%) except for leisure physical activity which is expressed as mean ± standard deviation. Only statistically significant differences are reported in the p columns.

**Table 8 nutrients-14-02493-t008:** Multivariate analysis of parameters associated with hepatocellular carcinoma (HCC).

Current Alcohol Consumption				LACU			
**Parameter**	**OR**	**95% CI**	** *p* **	**Parameter**	**OR**	**95% CI**	** *p* **
Male sex	5.75	1.68–19.61	0.005	Male sex	4.65	1.47–14.67	0.009
Age	1.12	1.05–1.18	<0.001				
Cirrhosis	7.96	2.79–22.76	<0.001	Cirrhosis	11.04	4.21–28.93	<0.001
BMI	1.00	0.90–1.12	0.998	BMI	0.96	0.86–1.08	0.529
Hypertension	3.47	0.85–14.20	0.083	Hypertension	6.12	1.60–23.39	0.008
Diabetes	1.98	0.65–6.01	0.227	Diabetes	2.26	0.82–6.26	0.116
**Alcohol intake**				**Alcohol intake**			
Abstinent	Reference			Abstinent	Reference		
C1	0.26	0.07–1.01	0.054	Q1–3	0.32	0.08–0.98	0.043
C2	0.78	0.21–2.91	0.705	Q4	1.79	0.54–5.92	0.342

## Data Availability

The data presented in this study are available on request from the corresponding author. The data are not publicly available due to privacy restrictions.

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
