# Peer review of "Very Low Alcohol Consumption Is Associated with Lower Prevalence of Cirrhosis and Hepatocellular Carcinoma in Patients with Non-Alcoholic Fatty Liver Disease"

_nutrients, 2022, doi:10.3390/nu14122493_

Round 1
Reviewer 1 Report
Major points.
- Clarify the etiology of cirrhosis. Alcohol or alcohol + steatosis
- Exclude patients with cirrhosis who have previously been admitted due to liver decompensation.
- To do all analysis excluding Binge-drinkers
- Table 5 and 7. The reference should be non-drinkers
- Say the follow-up time of the patients
- Explain because the transaminases are normal.
- Explain the exclusive consumers of beer and spirits. And to compare the risk of cirrhosis and HCC among nonalcoholic, exclusive wine consumers, exclusive beer consumers, and exclusive spirits consumers.
- How many hepatocellular carcinomas occur in patients without cirrhosis.
- Of all the binge-drinkers how many were for wine - beer or spirits.
- What was the concordance of the responses between the two interviews separated by 2 years?
- Alcohol consumption How long time have been consuming alcohol?
Author Response
- Clarify the etiology of cirrhosis. Alcohol or alcohol + steatosis.
ANSWER: please, see lines 72-77 in materials and methods: all patients fulfilled NAFLD diagnosis criteria, including the quantity of alcohol consumed (< 14 units per week for women and < 21 units per week for men).
- Exclude patients with cirrhosis who have previously been admitted due to liver decompensation.
ANSWER: none of the patients included in the present study had been previously admitted due to liver decompensation.
- To do all analysis excluding Binge-drinkers
ANSWER: please, see the supplementary Tables 1S-4S for the whole data on binge and non binge drinkers. As also patients that reported binge drinking episodes did not exceed the quantity considered for NAFLD diagnosis and reported above, we decided to consider them all together. We included a paragraph on the results section to illustrate binge drinkers characteristics (lines 304-318).
- Table 5 and 7. The reference should be non-drinkers.
ANSWER: we changed the tables accordingly to the reviewer's comments
- Say the follow-up time of the patients
ANSWER: there is no follow up, it is a picture of patients at their first visit in outpatients liver unit (lines 71-74 in the materials and methods section).
- Explain because the transaminases are normal.
ANSWER: in NAFLD, transaminases are reported to be normal in more than 25% of cases (in some reports this percentage reaches 80% of cases), and this does not mean than an underlying fibrosis condition can be excluded. Please, see for references:
- Browning JD et al. Hepatology 2004;40:1387-95.
- Sanyal D, et al. Indian J Endocrinol Metab 2015;19:597-601
- Ma X et al. BMC Gastroenterology 2020, 10. https://doi.org/10.1186/s12876-020-1165-z
In our cohort of patients, transaminases are altered in 53% of cases.
- Explain the exclusive consumers of beer and spirits. And to compare the risk of cirrhosis and HCC among nonalcoholic, exclusive wine consumers, exclusive beer consumers, and exclusive spirits consumers.
ANSWER: please, see the results section lines 184-188, 255-258, 332-376 including the added Table 4, 383-384, 417-419. Tables 2,3,5,7 were modified including data on the kind of alcoholic beverage consumed.
- How many hepatocellular carcinomas occur in patients without cirrhosis.
ANSWER: 10 HCC occurred in patients without cirrhosis (3 among the abstainers and 7 among the drinkers, all of them in Q4 group). Please, see the results section lines 415-416.
- Of all the binge-drinkers how many were for wine - beer or spirits.
ANSWER: please, see the supplementary Tables 1S-4S for the whole data on binge and non binge drinkers and lines 304-318 in the results section.
- What was the concordance of the responses between the two interviews separated by 2 years?
ANSWER: For the patients that we considered for the study, the concordance between the two questionnaires was excellent (100% for the kind of alcoholic beverages and Cohen k = 0.89 for the quantity).
- Alcohol consumption How long time have been consuming alcohol?
ANSWER: overall, the alcohol consuming time ranged from 0 to 60 years. We added a line in Tables 2,3,4,5,7 and in Supplementary tables 1-5 reporting this information.
Reviewer 2 Report
The typesetting of tables should be improved.
The normality of the chosen clients was not present in the manuscript
Author Response
- The typesetting of tables should be improved.
ANSWER: we modified the lineup of the tables (please see the manuscript and the supplementary material)
- The normality of the chosen clients was not present in the manuscript
ANSWER: we do not have a control population as we analyzed all patients coming to our NAFLD outpatient liver unit (please, see section materials and methods, lines 72-74). We have considered as controls our patients not drinking alcohol and we have re-done all the statistics using this cohort ad reference population (please, see Tables 6 and 8).
Round 2
Reviewer 1 Report
Nothing